# The Presence of Four Pathogenic Oral Bacterial Species in Six Wild Snake Species from Southern Taiwan: Associated Factors

**DOI:** 10.3390/microorganisms12020263

**Published:** 2024-01-26

**Authors:** Wen-Hao Lin, Tein-Shun Tsai, Po-Chun Chuang

**Affiliations:** 1Institute of Wildlife Conservation, National Pingtung University of Science and Technology, Pingtung 912301, Taiwan; chrislin840518@gmail.com; 2Department of Biological Science and Technology, National Pingtung University of Science and Technology, Pingtung 912301, Taiwan; 3Department of Computer Science and Engineering, National Sun Yat-sen University, Kaohsiung 804201, Taiwan; zhungboqun@gmail.com; 4Department of Emergency Medicine, Kaohsiung Chang Gung Memorial Hospital, Kaohsiung 833401, Taiwan

**Keywords:** diversity, venomous snake, microbe, pathogenesis, conservation medicine, human health

## Abstract

The oral cavity of snakes serves as a habitat for various microorganisms, some of which may include potential zoonotic pathogens posing risks to hosts and causing wound infections in snakebite victims. Clinical studies on snakebite cases in Taiwan have identified specific pathogens, such as *Enterococcus faecalis* (Gram-positive), *Morganella morganii*, *Aeromonas hydrophila*, and *Pseudomonas aeruginosa* (Gram-negative). However, the prevalence of these bacteria in the oral cavity of wild snakes remains largely unknown. This study investigated the occurrence of these bacteria in six wild snake species (*Naja atra*, *Bungarus multicinctus*, *Trimeresurus stejnegeri*, *Protobothrops mucrosquamatus*, *Boiga kraepelini*, and *Elaphe taeniura friesi*) from southern Taiwan, along with factors influencing their presence. Oropharyngeal swab samples were collected from a substantial number of wild-caught snakes (*n* = 1104), followed by DNA extraction, polymerase chain reaction, and gel electrophoresis. The band positions of samples were compared with positive and negative controls to determine the presence of target bacteria in each sample. The overall occurrence rates were 67.4% for *E. faecalis*, 31.5% for *M. morganii*, 8.2% for *A. hydrophila*, and 7.7% for *P. aeruginosa*. Among snake species, *B. kraepelini* exhibited dominance in *E. faecalis* (93.4%), *A. hydrophila* (17.1%), and *P. aeruginosa* (14.5%), while male *N. atra* showed dominance in *M. morganii* (51.3%). The occurrence of *E. faecalis* was lowest in winter. The results of multiple logistic regression analyses suggest that factors such as species, sex, temperature, season, and coexisting pathogens may have a significant impact on the occurrence of target bacteria. These findings have implications for wildlife medicine and snakebite management.

## 1. Introduction

Snakebites are categorized as neglected tropical diseases [1,2,3]. Annually, about 2.7 million people worldwide suffer from snakebites, leading to mortality cases ranging between 81,000 and 138,000 [4]. Snakebite-related deaths not only attribute fatalities to the venom’s toxicity but also to secondary infections caused by bacteria [5,6,7,8]. Numerous studies suggest that bacterial species isolated from snakebite wounds correspond to the oral flora of the responsible snake species [9,10]. However, wound contamination can also result from the victim’s skin tissue, pre-hospital environments, and inadequate wound management, introducing external bacterial strains unrelated to the snake’s oral flora [1,11,12,13,14].

Snake oral saliva contains a diverse bacterial community, including harmless environmental strains and potential pathogenic species harmful to humans [10,15,16]. Following a snakebite, the venom components may lead to necrosis of affected muscles and skin tissues, creating an environment conducive to the growth of bacteria carried by snake oral saliva [1,8,9,17,18]. Without proper attention and treatment, even bites from non-venomous snakes can result in bacterial invasion, leading to bacterial infections and the development of various chronic diseases, potentially resulting in death in severe cases [1,19,20,21]. Snake antivenom therapy may not prevent the subsequent symptoms caused by bacterial infections. Although antibiotics are commonly used to treat bacterial infections, administering them without understanding the specific infection source may be ineffective and contribute to antibiotic resistance, considering the varying sensitivity of bacterial species to antibiotics [1,10,22].

Targeted approaches offer precision and quantification for known target microbes, while non-targeted approaches provide a more comprehensive and unbiased exploration of the microbial landscape. Researchers should use a combination of both approaches to gain a deeper understanding of microbiological systems. In our investigation, we initially utilized a non-targeted approach to analyze the oral microbiota of seven species of wild venomous snakes in Taiwan, employing high-throughput amplicon sequencing of the full-length 16S rRNA gene [16]. Subsequently, we will conduct a targeted approach focusing on specific oral microbes using a molecular genotyping method. The bacterial species associated with secondary infections potentially resulting from snakebites are diverse and include aerobic Gram-positive bacteria, Gram-negative bacteria, and anaerobic bacteria (see Appendix A). Infections caused by Gram-positive bacteria can manifest with symptoms such as abscess formation, cellulitis, sepsis, meningitis, urinary tract infections, bacterial infections at the bite site, and ocular infections. *Enterococcus faecalis*, a common pathogenic Gram-positive bacterium found in the gastrointestinal tract, is frequently implicated. Most Gram-negative bacteria are pathogenic, and *Morganella morganii* is a major contributor to abscess formation and gastrointestinal bacterial infections. It is also frequently detected as a multi-drug-resistant bacterium carried by snakes in their oral cavity. *Pseudomonas aeruginosa* can lead to soft tissue infections in patients, including necrotizing cellulitis, folliculitis, dermatitis, urethritis, and osteomyelitis, and can induce immunosuppressive reactions in the snakebite site, potentially resulting in nosocomial infections. Infections caused by *Aeromonas hydrophila* can result in hemorrhagic sepsis and diseases related to diarrhea. Other Gram-negative bacteria can cause various infection symptoms such as gastroenteritis, sepsis, respiratory tract infections, meningitis, diarrhea, fever, and soft tissue infections [8,9,15,23,24,25,26,27,28]. The abundance of the aforementioned four species ranks high among the pathogens identified in our previous study [16].

The bacterial species carried by the oral flora of snakes can impact not only humans but also infect the host snakes themselves [21,26,29]. The oral cavity of reptiles may sustain wounds from struggling prey during feeding, tooth loss during hunting, or the ingestion of foreign substances such as artificial waste, strings, or fishing nets. These injuries can lead to swelling and inflammatory symptoms [30,31,32]. Inadequate control of environmental temperature and humidity in snake captivity can induce immune suppression in snakes, making them susceptible to the invasion of pathogenic bacteria through the respiratory and gastrointestinal tracts, resulting in systemic infections or even death [33,34,35]. Aerobic Gram-negative bacteria are generally considered the most pathogenic bacterial species for reptiles and are the dominant taxa detected in individuals with oral inflammation. Common genera include *Pseudomonas*, *Aeromonas*, *Proteus*, and *Escherichia* [31,33,36]. Among them, *Pseudomonas aeruginosa*, *Providencia rettgeri*, and *Stenotrophomonas maltophilia* are often dominant in the oral cavity of snakes with oral inflammation. *Aeromonas hydrophila* has historically exhibited the highest occurrence rate and is a significant zoonotic pathogen causing infections and death in snakes, commonly distributed in the oral cavity or respiratory tract of snakes [26,33,35,37,38]. To mitigate antibiotic resistance and enhance infection recovery rates, a comprehensive understanding of the carried bacterial species is essential for modifying subsequent treatment approaches and determining the appropriate types of antibiotics to administer [39].

The occurrence of bacterial species and their quantities may vary due to factors such as snake species, season, habitat, snake health, feeding strategies, and prey types [8,9,10,14,21,24,25,26,40]. Even closely related snake species may carry different oral bacterial species if they have different ranges of activity [10,14]. Many studies culture bacterial samples on agar plates before bacterial identification, but the cultured bacterial community may differ from the original community [21,41,42,43,44,45]. Non-cultivation methods can identify species that are not suitable for artificial cultivation [16,41].

Taiwan is home to six main venomous snake species: *Trimeresurus stejnegeri*, *Protobothrops mucrosquamatus*, *Naja atra*, *Bungarus multicinctus*, *Daboia siamensis*, and *Deinagkistrodon acutus* [46,47]. Annually, approximately 800 to 1000 people are bitten by venomous snakes in Taiwan [48,49]. Research on these venomous snakes often involves examining the clinical symptoms and wound tissue samples of snakebite patients to identify pathogenic bacterial species. Many previous studies focusing on snakebite patients in Taiwan have primarily used medical records from large hospitals in the northern or central regions for retrospective analysis [23,46,50], with only a few focusing on the southern region [49]. In wound samples from snakebite patients in Taiwan, Chen et al. [23] and Huang et al. [46] found that *M. morganii* was the most prevalent among Gram-negative bacteria, followed by *A. hydrophila*; *Enterococcus* strains were the most common among Gram-positive bacteria. Chung et al. [49] analyzed the bacterial composition of six major venomous snake oral samples by the culture method, revealing that the most abundant species among Gram-positive bacteria was *E. faecalis*, while *P. aeruginosa* and *Proteus vulgaris* were the most common among Gram-negative bacteria. Further studies, without cultivating bacteria from oral samples, are necessary to comprehend the occurrence of pathogenic bacteria in the oral cavity of Taiwanese snakes. Additionally, there is still limited exploration of the associations between bacterial species and factors such as snake species, region, season, and the body condition of snakes. Moreover, there is a lack of research on the oral bacterial species in non-venomous snakes in Taiwan, which also harbor potential pathogenic bacteria.

This study aims to investigate the occurrence of pathogenic zoonotic bacterial species in the oral cavities of commonly encountered wild snake species in Kaohsiung, southern Taiwan. The targeted bacterial species include *E. faecalis*, a Gram-positive bacterium, as well as *M. morganii*, *P. aeruginosa*, and *A. hydrophila*, which are Gram-negative bacteria. Polymerase chain reaction (PCR) technology is employed to detect the presence of these four pathogenic microorganisms. Multiple logistic regression analyses are utilized to examine the association between the occurrence of the four bacterial species and various factors, including snake species, sex, snout-vent length, body mass, season, duration between capture and sampling, temperature and rainfall, altitude, artificial ground area around the capture site, and coexisting bacterial species. The findings of this study can serve as a foundation for antibiotic sensitivity testing of bacterial species and facilitate the development of veterinary medicine for wild snakes in Taiwan. This information can, therefore, be valuable in understanding the treatment approaches for secondary infections in humans suffering from snakebites or bacterial infections, whether in captive or wild snakes, stemming from oral microbes.

## 2. Materials and Methods

### 2.1. Snake Collection

From September 2021 (early autumn) to November 2022 (late autumn), we conducted the collection of six common wild snake species in various administrative districts of Kaohsiung City. The collection comprised four venomous species (*N. atra*, *B. multicinctus*, *T. stejnegeri*, and *P. mucrosquamatus*), one mildly venomous species (*Boiga kraepelini*), and one non-venomous species (*Elaphe taeniura friesi*) (Figure 1). The majority of captured snakes were transported to the animal house at the National Pingtung University of Science and Technology for sample collection and morphological measurements within one to two weeks after capture. Before sampling, continuous access to water was provided, with no administration of food or medication [9].

### 2.2. Oropharyngeal Sample Collection

Oropharyngeal swab samples were collected from each snake using two cotton swabs [9]. Sampling was avoided under the following conditions: excessive blood due to injury in the oral cavity, severe ulceration and purulent discharge from old wounds, inability to ensure the cotton swab’s uncontaminated status during sampling, extremely small body size, excessive parasitism in the oral region, poor health condition, or imminent death of the snake [9]. A disposable sterile tongue depressor (with a hole in the middle) was used as a mouth opener during sampling (see Figure 1h in [16]) to prevent cross-contamination. After opening the mouth, sterile cotton swabs were employed to sample from the oropharyngeal area. The swabs were rotated to ensure even adherence of saliva to the cotton [9]. Following sampling, the sample swabs were placed in a sterilized 1.5 mL microcentrifuge tube and stored at −80 °C.

### 2.3. DNA Extraction, PCR Amplification, and Electrophoresis

The DNA from oropharyngeal saliva samples was extracted through organic extraction (chloroform:isoamyl alcohol = 24:1) [51], and DNA concentration was determined using a Nano-300 Micro-Spectrophotomer (Medclub Scientific, Taoyuan, Taiwan). The final volume of the PCR reaction was 25 μL, with the reagent composition including 15 μL sterile distilled water, 2.5 μL of 10× Taq PCR buffer containing 15 mM MgCl (Cyrusbioscience, New Taipei, Taiwan), 2 μL of 10 μM forward and reverse primers (Purigo Biotech, Taipei, Taiwan) targeting the 16S rRNA gene segment, 0.5 μL of 10 mM dNTPs (ABclonal, Woburn, MA, USA), 2.5 U of Taq DNA Polymerase (Cyrusbioscience, New Taipei, Taiwan), and 2.5 μL of DNA sample. The primer selection and temperature-time configuration of PCR specific to each bacterial species are detailed as follows:

#### 2.3.1. *E. faecalis*

The forward primer sequence was 5′-GTT TAT GCC GCA TGG CAT AAG AG-3′ (located at base position 167–189 of the *E. faecalis* 16S rRNA, GenBank accession no. NR_113902), and the reverse primer sequence was 5′-CCG TCA GGG GAC GTT CAG-3′ (located at base position 459–476 of the *E. faecalis* 16S rRNA, GenBank accession no. NR_113902). The PCR protocol consisted of an initial denaturation step at 95 °C for 2 min, followed by 26 cycles of denaturation at 95 °C for 30 s, annealing at 60 °C for 1 min, and extension at 72 °C for 1 min. The final extension step was performed at 72 °C for 2 min. The expected product size after gel electrophoresis was 310 base pairs (Figure 2A) [52].

#### 2.3.2. *M. morganii*

The forward primer sequence was 5′-TTT CAG TCG GGA GGA AGG TG-3′ (located at base position 357–376 of the *M. morganii* 16S rRNA, GenBank accession no. NR_043751), and the reverse primer sequence was 5′-GGG GAT TTC ACA TCT GAC TC-3′ (located at base position 515–534 of the *M. morganii* 16S rRNA, GenBank accession no. NR_043751). The PCR protocol consisted of an initial denaturation step at 95 °C for 2 min, followed by 30 cycles of denaturation at 94 °C for 1 min, annealing at 65 °C for 45 s, and extension at 72 °C for 2 min. The final extension step was performed at 72 °C for 7 min. The expected product size after gel electrophoresis was 178 base pairs (Figure 2B) [53].

#### 2.3.3. *P. aeruginosa*

The forward primer sequence was 5′-GGG GGA TCT TCG GAC CTC A-3′ (located at base position 185–203 of the *P. aeruginosa* 16S rRNA, GenBank accession no. NR_117678), and the reverse primer sequence was 5′-TCC TTA GAG TGC CCA CCC G-3′ (located at base position 1122–1140 of the *P. aeruginosa* 16S rRNA, GenBank accession no. NR_117678). The PCR protocol included an initial denaturation step at 95 °C for 2 min, followed by 25 cycles of denaturation at 94 °C for 20 s, annealing at 58 °C for 20 s, and extension at 72 °C for 40 s. The final extension step was performed at 72 °C for 1 min. The expected product size after gel electrophoresis was 956 base pairs (Figure 2C) [54].

#### 2.3.4. *A. hydrophila*

The forward primer sequence was 5′-GAA AGG TTG ATG CCT AAT ACG TA-3′ (located at base position 445–467 of the *A. hydrophila* 16S rRNA, GenBank accession no. NR_074841), and the reverse primer sequence was 5′-CGT GCT GGC AAC AAA GGA CAG-3′ (located at base position 1110–1130 of the *A. hydrophila* 16S rRNA, GenBank accession no. NR_074841). The PCR protocol included an initial denaturation step at 95 °C for 5 min, followed by 30 cycles of denaturation at 94 °C for 2 min, annealing at 56 °C for 2 min, and extension at 72 °C for 2 min. The final extension step was performed at 72 °C for 10 min. The expected product size after gel electrophoresis was 685 base pairs (Figure 2D) [55].

#### 2.3.5. Positive and Negative Controls

Four strains were included in a set of positive and negative control groups during the PCR process. The positive control consisted of a DNA sample from a specific strain of bacteria that had been cultured and purified. After gel electrophoresis and sequencing, the bacterial sequences were verified using the NCBI BLAST (*E. faecalis* NBRC 100481, *M. morganii* DSM 14850, *P. aeruginosa* DSM 50071, and *A. hydrophila* ATCC 7966). The negative control was sterile distilled water to confirm the absence of contamination [52,55].

#### 2.3.6. Agarose Gel Electrophoresis

The PCR products were separated using a 1.5% agarose gel. For each sample, 4 μL of the PCR product was mixed with 1 μL of 5× prestained loading dye. Electrophoresis was conducted in 1% TAE buffer at 110 volts for 30 min. A 100 bp-prestained DNA ladder was used for size comparison [54]. Following electrophoresis, the gel was exposed to UVA light, and images were captured. The DNA ladder and the positive control served as reference points to determine whether bands corresponding to the PCR products of the target bacterial strains appeared at the expected positions and sizes. This method was employed to assess the presence of target bacterial DNA fragments (Figure 2). Additionally, we sequenced the PCR products of select samples to validate the accuracy of species identification by referring to NCBI BLAST.

### 2.4. Statistical Methods

Contingency table analyses, including the *X*^2^ test and Fisher’s exact test, were performed using PAST (version 4.12) to examine the significance of associations between variables. In cases where a significant association (*p* < 0.05) was identified, post-hoc analyses were conducted using the Bonferroni correction to calculate an adjusted *p* value for each cell in the contingency table. This adjustment aimed to assess the significance of each variable’s contribution to deviating from the expected values under the null hypothesis of no association.

Furthermore, multiple logistic regression was employed to investigate the association between bacterial occurrence and morphometric or environmental factors in the sampled snakes [56]. The categorical factors include snake species, sex, capture season, and coexistence of other bacterial species; the numerical factors include snout-vent length, scaled mass index, artificial ground area within a 100 m radius of the capture site, average temperature on the capture day, average rainfall for the capture month, elevation of the capture site, and the duration between capture and sampling. The scaled mass index was calculated as follows [57,58]:$$Scaled\;mass\;index={M_{i}(\frac{L_{0}}{L_{i}})}^{b_{SMA}}$$
The b_SMA_ is the slope for the bivariate linear relationship between log_10_(body mass) and log_10_(snout-vent length) following the standardized major axis regression analysis, which was performed using RStudio (version 2022.07.2+576). L_0_ represents the mean snout-vent length of all snakes, L_i_ is the snout-vent length of an individual snake, and M_i_ is the body mass of an individual snake. The capture season is differentiated by months, with spring encompassing March to May, summer from June to August, autumn from September to November, and winter from December to February [59]. Temperature and rainfall data were sourced from the nearest automatic weather station to the capture site, provided by the Central Weather Bureau’s Observation Data Inquire Service (CODiS; https://codis.cwa.gov.tw/, accessed on 15 January, 2023). Land use types at capture sites were obtained from the National Land Surveying and Mapping Center (Taiwan MAP service; https://maps.nlsc.gov.tw/, accessed on 15 January, 2023), including paddy fields, dry fields, orchards, rivers, lakes, sea areas, and forests as non-artificial ground, with all others categorized as artificial ground. Considering the daily movement distances of snakes [60], the artificial ground area within a 100 m radius around the capture point was measured using ArcGIS (version 10.8.1).

Multiple logistic regression model analyses were conducted using JMP (version 10.0.0) and IBM SPSS Statistics (version 29.0.0.0), evaluating the impact of all morphometric and environmental factors on the occurrence of target bacterial species and establishing the full model. A reduced model was formed by including only factors with significant impact (*p* < 0.05). If the AICc (corrected Akaike information criterion) value of the reduced model was smaller than that of the full model, the reduced model was considered the most parsimonious regression model. Examination of the Exp(B) values (i.e., odds ratio) for each factor was conducted, where values greater than 1 indicated a positive association with bacterial occurrence and values less than 1 indicated a negative association. Additionally, the significance of interactions between factors was also tested, and no significant interactions were found.

## 3. Results

### 3.1. Sampled Snake Quantity by Sex

Between 1 September 2021, and 30 November 2022, a total of 1104 snakes were captured and sampled, including 354 *N. atra* (54.5% males, 45.4% females), 159 *B. multicinctus* (60.3% males, 39.6% females), 216 *T. stejnegeri* (40.7% males, 59.2% females), 118 *P. mucrosquamatus* (40.6% males, 59.3% females), 76 *B. kraepelini* (61.8% males, 38.2% females), and 181 *E. t. friesi* (45.3% males, 54.6% females).

### 3.2. Comparison of Bacterial Occurrence by Season, Snake Species, and Sex

#### 3.2.1. *E. faecalis*

*E. faecalis* was detected in a total of 744 snakes; the overall occurrence rates were 67.4%. There was a significant difference in the occurrence rates of *E. faecalis* among seasons (*X*^2^ = 55.6, *d.f.* = 4, *p* < 0.000001; Figure 3); the lowest occurrence rate occurred in winter 2021 (44.4%), while the highest was observed in autumn 2022 (80.4%). Significant differences were also found in the occurrence rates among snake species (*X*^2^ = 142, *d.f.* = 5, *p* < 0.000001; Figure 4). *B. kraepelini* (93.4%), *E. t. friesi* (84.5%), and *B. multicinctus* (83.0%) exhibited significantly higher occurrence rates compared to *P. mucrosquamatus* (50.0%) and *T. stejnegeri* (42.6%). There was no significant difference in the overall occurrence rates between all males (69.7%) and all females (65.1%) (Fisher’s exact test; *p* = 0.109; Figure 5); no significant differences were observed in the occurrence rates between males and females within each snake species (Fisher’s exact test; all adjusted *P*s > 0.05).

#### 3.2.2. *M. morganii*

*M. morganii* was detected in a total of 348 snakes; the overall occurrence rates were 31.5%. There was a significant difference in the occurrence rates of *M. morganii* among seasons (*X*^2^ = 40.5, *d.f.* = 4, *p* < 0.000001; Figure 3); the lowest occurrence rate occurred in summer 2022 (20.9%), while the highest was observed in autumn 2022 (45.4%). Significant differences were also found in the occurrence rates among snake species (*X*^2^ = 83.9, *d.f.* = 5, *p* < 0.000001; Figure 4). *N. atra* (44.1%) and *T. stejnegeri* (42.1%) exhibited significantly higher occurrence rates compared to *E. f. taeniura* (12.7%) and *P. mucrosquamatus* (19.5%). The overall occurrence rate in males (35.0%) was slightly higher than in females (28.0%) (Fisher’s exact test; *p* = 0.0138; Figure 5). Specifically, the occurrence rate of *M. morganii* in male *N. atra* (51.3%) and *B. multicinctus* (35.4%) was higher than in females (35.4% and 12.7%, respectively) (Fisher’s exact test; adjusted *P*s < 0.05), while no significant sex differences were observed in the other snake species.

#### 3.2.3. *P. aeruginosa*

*P. aeruginosa* was detected in a total of 85 snakes; the overall occurrence rates were 7.7%. There was a significant difference in the occurrence rates of *P. aeruginosa* among seasons (*X*^2^ = 29.2, *d.f.* = 4, *p* < 0.00001; Figure 3); the lowest occurrence rate occurred in summer 2022 (3.2%), while the highest was observed in autumn 2022 (14.6%). *B. kraepelini* (14.5%) and *B. multicinctus* (14.5%) had higher occurrence rates compared to other species (*X*^2^ = 20.4, *d.f.* = 5, *p* = 0.00104; Figure 4). The overall occurrence rate in male snakes (8.5%) was not significantly different from that in females (6.9%) (Fisher’s exact test; *p* = 0.367; Figure 5); no significant differences were observed in the occurrence rates between males and females within each snake species (Fisher’s exact test; all adjusted *P*s > 0.05).

#### 3.2.4. *A. hydrophila*

*A. hydrophila* was detected in a total of 90 snakes; the overall occurrence rates were 8.2%. There was no significant difference in the occurrence rates of *A. hydrophila* among seasons (*X*^2^ = 2.81, *d.f.* = 4, *p* = 0.590; Figure 3). However, there was a significant difference in the occurrence rates among snake species (*X*^2^ = 29.9, *d.f.* = 5, *p* < 0.00001; Figure 4), with *B. kraepelini* (17.1%) and *B. multicinctus* (16.4%) exhibiting significantly higher occurrence rates. The overall occurrence rate in males (8.7%) was not significantly different from that in females (7.6%) (Fisher’s exact test; *p* = 0.583; Figure 5); no significant differences were observed in the occurrence rates between males and females within each snake species (Fisher’s exact test; all adjusted *P*s > 0.05).

### 3.3. Association Factors of Bacterial Occurrence

The full model of multiple logistic regression incorporated both categorical and numerical factors. Table 1 displays the range of values for the numerical factors.

#### 3.3.1. *E. faecalis*

Factors significantly associated with the occurrence of *E. faecalis* included snake species, temperature, elevation, season, and coexisting bacterial species (Table 2). Using *B. kraepelini* as a reference, *N. atra*, *B. multicinctus*, *T. stejnegeri*, and *P. mucrosquamatus* all exhibited a significantly lower occurrence of *E. faecalis* (odds ratio < 1). Temperature and elevation showed a significant positive association with bacterial occurrence (odds ratio > 1), but the effect of elevation is actually minor because the B value for elevation is close to 0 (Table 2). In comparison to winter 2021, the occurrence rate in spring 2022 was significantly lower. The presence of the other target bacterial species demonstrated a significant positive association with the occurrence rate of *E. faecalis*.

#### 3.3.2. *M. morganii*

Factors significantly associated with the occurrence of *M. morganii* included snake species, sex, temperature, season, and coexisting bacterial species (Table 3). Using *B. kraepelini* as a reference, *N. atra* and *T. stejnegeri* exhibited a significantly higher occurrence of *M. morganii*. Compared to males, the occurrence rate in females was significantly lower. Temperature showed a significant negative association with occurrence. The presence of the other target bacterial species demonstrated a significant positive association with the occurrence rate of *M. morganii*.

#### 3.3.3. *P. aeruginosa*

Factors significantly associated with the occurrence of *P. aeruginosa* included only seasonal and coexisting bacterial species (Table 4). Compared to winter 2021, the occurrence rate in summer 2022 was significantly lower. The presence of the other target bacterial species demonstrated a significant positive association with the occurrence rate of *P. aeruginosa*.

#### 3.3.4. *A. hydrophila*

Factors significantly associated with the occurrence of *A. hydrophila* included only snake species and coexisting bacterial species (Table 5). Using *B. kraepelini* as a reference, *T. stejnegeri*, *N. atra*, and *E. t. friesi* showed a significantly lower occurrence of *A. hydrophila*. The presence of the other target bacterial species demonstrated a significant positive association with the occurrence rate of *A. hydrophila*.

## 4. Discussion

The factors contributing to bacterial growth in a snake’s oral cavity, aside from venom, may encompass the balance of oral microbiota, the type of prey, environmental factors (humidity, temperature, and substrate composition), the immune system, oral hygiene, and the presence of injury or infection [8,9,10,14,21,24,25,26,40]. This study demonstrates significant associations between the occurrence rates of the four bacterial species and various factors, as summarized in Table 6. Notably, certain factors like sex, snout-vent length, scaled mass index, artificial ground area, rainfall, and the duration between capture and sampling did not exhibit a significant association with bacterial occurrences. In contrast, the presence of other bacterial species in the oral cavity consistently exhibited a positive association with the occurrence rates of the target bacteria, given that sampling of snakes with compromised health conditions or immune activity was avoided in this study. This finding suggests that pathogenic bacterial species may share growth environments with each other, as seen in previous research [61]. It also implies that, whether dealing with secondary infections in humans following snake bites or bacterial infections in captive or wild snakes due to oral wounds, multiple pathogenic sources may contribute to the infections. Lin et al. [62] and Yeh et al. [63] indicated that *M. morganii* and *E. faecalis* were the most common sources of polymicrobial infections. Both *E. faecalis* and *M. morganii* can lead to bacteremia, while *A. hydrophila* and *P. aeruginosa* may result in urinary tract infections [23,46]. The presence of multiple pathogens in infections implies varying degrees of antibiotic resistance, underscoring the importance of thorough evaluation and testing of antibiotic sensitivity in the future. The associated factors for the occurrence rates of bacterial species are discussed as follows:

Some studies suggest that snake venom possesses antibacterial properties [64,65,66,67,68], hypothesizing that the venom can protect venomous snakes from bacterial infections carried by prey during feeding. However, conflicting research supports the notion that venomous snakes display a greater diversity of oral bacteria compared to non-venomous ones [21,69]. In the context of this study, pit vipers (*P. mucrosquamatus* and *T. stejnegeri*) exhibit a significantly lower occurrence rate of *E. faecalis*, a bacterium whose growth has been demonstrated to be inhibited by venom from vipers and cobras [70]. In contrast, mildly venomous *B. kraepelini* and venomous *B. multicinctus* demonstrate a high occurrence rate of *E. faecalis*, *P. aeruginosa*, and *A. hydrophila*. Furthermore, the occurrence rate of *M. morganii* in the oral cavity of venomous *N. atra* and *T. stejnegeri* surpasses that in other snake species, consistent with prior research on patient wounds [71] and the oral cavities of snakes [16]. The reasons behind the heightened bacterial occurrence in the above (mildly) venomous species warrant further investigation. Whether the antibacterial properties of venom are related to only specific venom compositions also remains to be confirmed.

Our study reveals a positive correlation between the occurrence rate of *E. faecalis* and environmental temperature, aligning with previous findings that *E. faecalis* thrives in warmer temperatures [72]. Since *E. faecalis* predominantly inhabits the gastrointestinal tract of animals, the oral microbiota of snakes is influenced by the excretion of prey during feeding [73]. The increased activity of snakes due to higher environmental temperatures may lead to an elevated frequency of predation [60], indirectly contributing to the higher occurrence rate of *E. faecalis*.

*M. morganii* exhibits the highest occurrence rate among Gram-negative bacteria in this study. This observation aligns with findings from both clinical research on patient wounds and snake oral microbiota studies [23,46,50]. Bacteriomic analysis reveals that *M. morganii* ranks among the 10 most abundant bacterial species in oropharyngeal samples of both *N. atra* and *T. stejnegeri* [16]. *M. morganii*, known to inhabit the gastrointestinal tract of animals, is also widely distributed in the natural environment, including soil, water sources, and sewage [74,75]. The higher occurrence rate of *M. morganii* in the oral cavity of male *N. atra* and *B. multicinctus* may be attributed to the larger activity range seen in males of some snake species [76,77], increasing the likelihood of contact with pathogens.

*P. aeruginosa* primarily inhabits human feces and sewage, with less occurrence in natural environments such as soil and streams [78]. This study exhibited the lowest occurrence rate among bacterial species, contrary to previous research on Taiwanese snake oral bacteria that used culture-based methods, identifying *P. aeruginosa* as the Gram-negative species with the highest occurrence rate [49]. This discrepancy may stem from the limitation that microbial compositions cultured on agar plates may not entirely represent the original oral microbiota, as indicated by several studies [21,41,42,43,44,45]. Studies have shown that *P. aeruginosa* outbreaks are more extensive in hospital areas than in non-hospital settings [78]. This suggests that the *P. aeruginosa* identified in clinical studies, particularly in patients with wound infections [23], may predominantly originate from nosocomial infections. Additionally, previous research investigating the seasonal association of *P. aeruginosa* using human samples demonstrated a peak in its occurrence during cooler months (October to March) [79], which aligns with our findings indicating a higher occurrence rate of *P. aeruginosa* in winter.

This study has several limitations: (1) The investigation did not analyze the impact of dietary factors on the incidence rates of bacteria in the sampled snake species due to their complex and overlapping feeding habits. Subsequent research could concentrate on snake species with simpler and non-overlapping diets to ascertain whether snake feeding habits significantly influence bacterial occurrence rates. (2) Previous studies [31,33,36] have identified Gram-negative bacteria, particularly *P. aeruginosa*, *P. rettgeri*, and *S. maltophilia*, as dominant strains in the oral cavity of snakes with oral inflammation. However, our study did not sample the oral bacterial community of unhealthy or sick snakes. Future studies should include samples from snakes exhibiting infectious symptoms to explore variations in oral microbiomes under different health conditions. (3) The bacterial strains selected for this study were aerobic or facultatively anaerobic. Future research on the oral bacterial communities of wild snakes should also consider investigating other types of pathogenic bacteria, including anaerobic strains such as *B. fragilis* and *Clostridium* spp.

## 5. Conclusions

Our research, involving extensive sampling of wild snakes in southern Taiwan, provides insights into the complex factors influencing the presence of oral bacteria in snake populations. Specifically, the prevalence of oral cavity bacteria ranked highest for *E. faecalis*, followed by *M. morganii*, *A. hydrophila*, and *P. aeruginosa*. Among snake species, *B. kraepelini* and *B. multicinctus* dominated in *E. faecalis*, *A. hydrophila*, and *P. aeruginosa*, while male *N. atra* and *T. stejnegeri* led in *M. morgani*. Winter exhibited the lowest occurrence of *E. faecalis*. Multiple logistic regression analyses underscored species, sex, temperature, elevation, season, and coexisting bacteria as potentially influential factors. The presence of other bacterial species in the oral cavity consistently showed a positive association with the occurrence rates of the target bacteria. While numerous studies have explored bacterial communities in the wounds of patients bitten by vipers and cobras or within the oral cavities of these snakes [8,23,44,46,80], research on other (mildly) venomous [81] snakes as well as non-venomous [21] snakes are notably scarce. To enhance veterinary therapy for such snakes, as well as snakes under different health conditions, additional studies on the bacterial community harbored within these snakes are deemed necessary. Our findings hold significance for both human clinical treatment and wild snake medicine efforts.

## Figures and Tables

**Figure 1 microorganisms-12-00263-f001:**
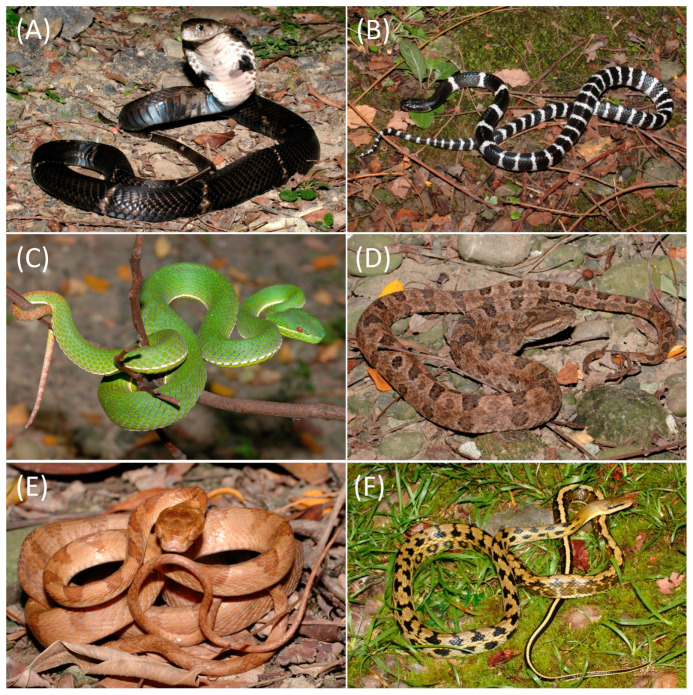
Snakes used for oropharyngeal sample collection: (**A**) *Naja atra*; (**B**) *Bungarus multicinctus*; (**C**) *Trimeresurus stejnegeri*; (**D**) *Protobothrops mucrosquamatus*; (**E**) *Boiga kraelipelini*; (**F**) *Elaphe taeniura friesi*. (Photos by Tein-Shun Tsai).

**Figure 2 microorganisms-12-00263-f002:**
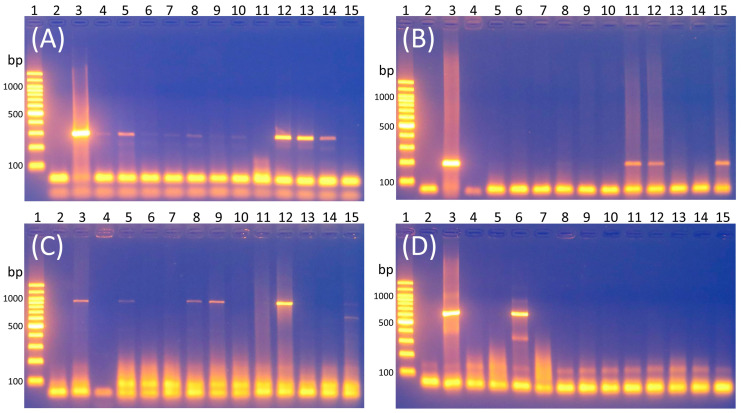
Photo examples of gel electrophoresis products illustrating the presence of (**A**) *Enterococcus faecalis*, (**B**) *Morganella morganii*, (**C**) *Pseudomonas aeruginosa*, and (**D**) *Aeromonas hydrophila* in oral samples of snakes. The lanes are as follows: Lane 1: 100-bp DNA ladder; Lane 2: negative control; Lane 3: positive control with a band size of 310, 178, 956, and 685 base pairs for each bacterial species, respectively; Lanes 4−15: samples.

**Figure 3 microorganisms-12-00263-f003:**
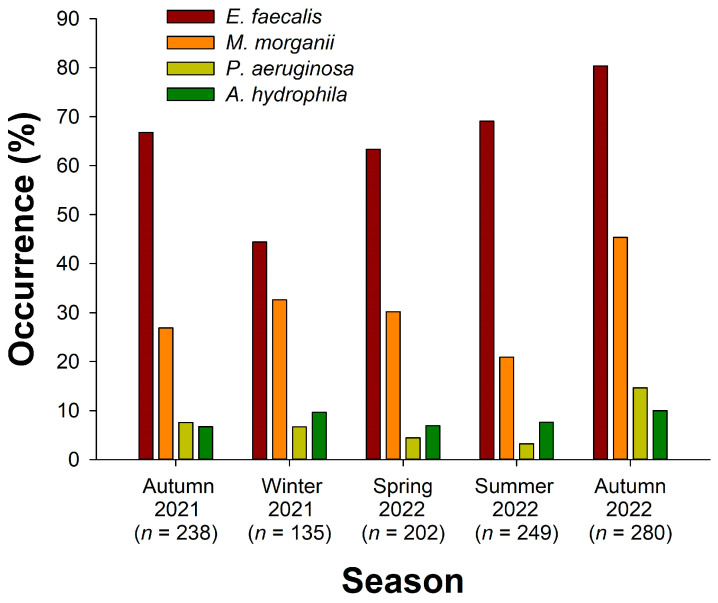
Comparison of the occurrence rates of four bacteria species (*E. faecalis*, *M. morganii*, *P. aeruginosa*, and *A. hydrophila*) in the oropharyngeal samples of snakes during different seasons (Autumn 2021, Winter 2021, Spring 2022, Summer 2022, and Autumn 2022).

**Figure 4 microorganisms-12-00263-f004:**
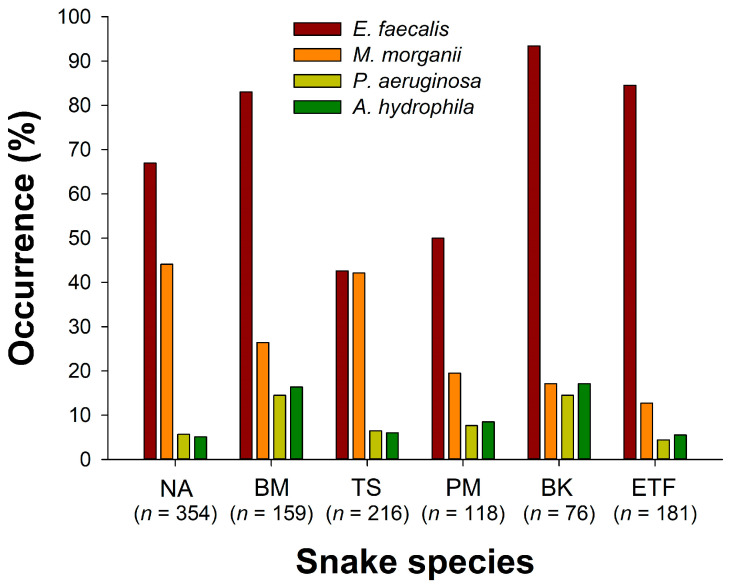
Comparison of the occurrence rates of four bacteria (*E. faecalis*, *M. morganii*, *P. aeruginosa*, and *A. hydrophila*) in the oropharyngeal samples of six snake species (NA: *N. atra*; BM: *B. multicinctus*; TS: *T. stejnegeri*; PM: *P. mucrosquamatus*; BK: *B. kraepelini*; ETF: *E. t. fries*).

**Figure 5 microorganisms-12-00263-f005:**
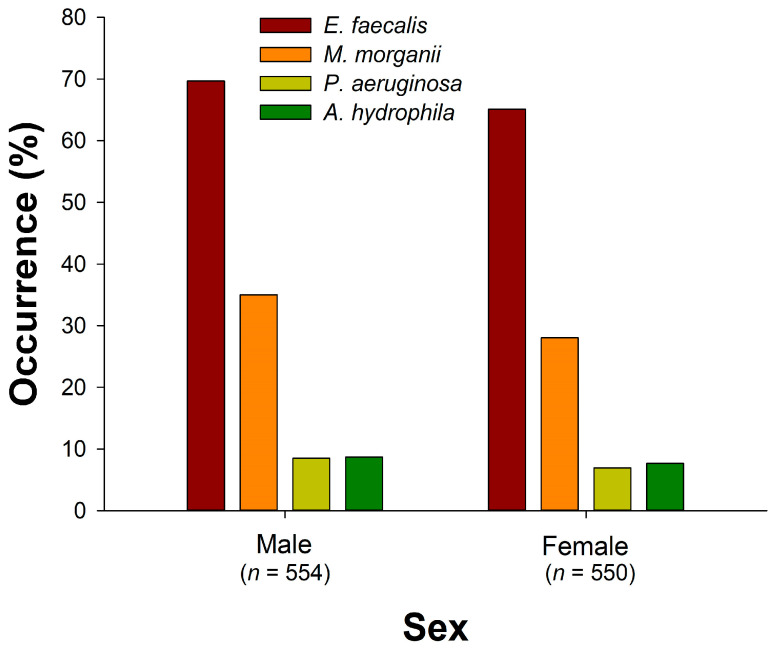
Sex-based comparison of the occurrence rates of four bacteria (*E. faecalis*, *M. morganii*, *P. aeruginosa*, and *A. hydrophila*) in the oropharyngeal samples of snakes.

**Table 1 microorganisms-12-00263-t001:** Minimum and maximum values of various numerical factors included in the full model of multiple logistic regression.

Factor	*B. kraepelini*		*B. multicinctus*		*N. atra*		*E. t. freisi*		*P. mucrosquamatus*		*T. stejnegeri*	
	Min	Max	Min	Max	Min	Max	Min	Max	Min	Max	Min	Max
Snout-vent length (cm)	29.8	120	21.3	126	29.5	135.2	32.9	192	21.3	92.6	21.1	73
Scaled mass index ^a^	26.18	819.43	44.66	7315.94	30.88	11,762.84	12.61	10,798.92	59.58	878.69	67.51	545.94
Artificial ground area (m^2^) ^b^	5128.98	31,415.96	2501.6	31,415.96	6943.17	31,415.96	4838.05	31,415.96	2557.07	31,415.96	4333.86	31,415.96
Temperature (°C)	19.1	31.3	18.6	31.3	17.7	31.6	15	31.5	15.3	30.3	15.8	31.3
Rainfall (mm)	0	13.3	0	12.6	0	15.7	0	14.3	0	7.8	0	18
Elevation (m)	4	430	3	320	1	320	3	252	7	248	2	283
Duration (days) ^c^	1	26	0	22	2	25	0	26	2	27	0	21

^a^ Refer to the text for explanations. ^b^ Artificial ground area within a radius of 100 m from the capture points. ^c^ Duration between capture and sampling.

**Table 2 microorganisms-12-00263-t002:** Association statistics for the multiple logistic regression between *E. faecalis* occurrence and various factors. The table includes the estimated coefficients (B), standard errors (S.E.), Wald statistics (Wald), degrees of freedom (*d.f.*), probabilities under null hypothesis (*P*), odds ratios (Exp(B)), and 95% confidence intervals (95% C.I.) of Exp(B). Only significant factors (*p* < 0.05) are presented. The reference category for subclass factors is indicated within parentheses following the specific factor name.

Factor	B	S.E.	Wald	*d.f.*	*p*	Exp(B)	95% C.I.	
							Lower	Upper
Snake species (*B. kraepelini*)			129.590	5	<0.001			
* N. atra*	−1.966	0.500	15.436	1	<0.001	0.140	0.052	0.373
* B. multicinctus*	−1.092	0.528	4.283	1	0.038	0.335	0.119	0.944
* T. stejnegeri*	−3.284	0.508	41.800	1	<0.001	0.037	0.014	0.101
* P. mucrosquamatus*	−2.582	0.522	24.480	1	<0.001	0.076	0.027	0.210
* E. t. friesi*	−0.500	0.520	0.925	1	0.336	0.607	0.219	1.680
Temperature	0.128	0.039	10.729	1	0.001	1.137	1.053	1.227
Elevation	0.007	0.003	6.915	1	0.009	1.007	1.002	1.012
Seasons (Winter 2021)			20.033	4	<0.001			
Autumn 2021	−0.203	0.365	0.311	1	0.577	0.816	0.399	1.668
Spring 2022	−0.618	0.361	2.940	1	0.086	0.539	0.266	1.093
Summer 2022	−0.657	0.445	2.175	1	0.140	0.518	0.217	1.241
Autumn 2022	0.332	0.375	0.786	1	0.375	1.394	0.669	2.906
Presence of *M. morganii*	1.549	0.192	65.060	1	<0.001	4.709	3.231	6.862
Presence of *A. hydrophila*	0.950	0.414	5.266	1	0.022	2.585	1.149	5.819
Presence of *P. aeruginosa*	1.836	0.547	11.277	1	<0.001	6.272	2.148	18.314

**Table 3 microorganisms-12-00263-t003:** Association statistics for the multiple logistic regression between *M. morganii* occurrence and various factors. Refer to the legend in Table 2 for an explanation of statistical symbols.

Factor	B	S.E.	Wald	*d.f.*	*P*	Exp(B)	95% C.I.	
							Lower	Upper
Snake species (*B. kraepelini*)			120.133	5	<0.001			
* N. atra*	2.164	0.353	37.496	1	<0.001	8.702	4.354	17.393
* B. multicinctus*	0.636	0.378	2.829	1	0.093	1.888	0.900	3.961
* T. stejnegeri*	2.184	0.379	33.301	1	<0.001	8.885	4.231	18.659
* P. mucrosquamatus*	0.716	0.422	2.878	1	0.090	2.047	0.895	4.683
* E. t. friesi*	0.054	0.400	0.018	1	0.892	1.056	0.482	2.310
Sex-Females (Males)	−0.365	0.150	5.887	1	0.015	0.694	0.517	0.932
Temperature	−0.087	0.038	5.205	1	0.023	0.917	0.851	0.988
Seasons (Winter 2021)			16.695	4	0.002			
Autumn 2021	−0.027	0.366	0.005	1	0.942	0.974	0.475	1.996
Spring 2022	0.100	0.353	0.080	1	0.777	1.105	0.553	2.210
Summer 2022	−0.155	0.438	0.125	1	0.723	0.857	0.363	2.020
Autumn 2022	0.631	0.357	3.119	1	0.077	1.880	0.933	3.789
Presence of *E. faecalis*	1.534	0.190	64.990	1	<0.001	4.639	3.194	6.737
Presence of *A. hydrophila*	1.156	0.264	19.099	1	<0.001	3.177	1.892	5.335
Presence of *P. aeruginosa*	0.741	0.269	7.595	1	0.006	2.099	1.239	3.556

**Table 4 microorganisms-12-00263-t004:** Association statistics for the multiple logistic regression between *P. aeruginosa* occurrence and various factors. Refer to the legend in Table 2 for an explanation of statistical symbols.

Factor	B	S.E.	Wald	*d.f.*	*p*	Exp(B)	95% C.I.	
							Lower	Upper
Seasons (Winter 2021)			19.347	4	<0.001			
Autumn 2021	−0.133	0.458	0.084	1	0.771	0.875	0.357	2.149
Spring 2022	−0.731	0.516	2.009	1	0.156	0.481	0.175	1.323
Summer 2022	0.527	0.419	1.584	1	0.208	1.693	0.746	3.846
Autumn 2022	−1.093	0.527	4.295	1	0.038	0.335	0.119	0.942
Presence of *E. faecalis*	2.045	0.529	14.943	1	<0.001	7.730	2.741	21.802
Presence of *M. morganii*	0.520	0.250	4.317	1	0.038	1.683	1.030	2.749
Presence of *A. hydrophila*	1.503	0.292	26.520	1	<0.001	4.496	2.537	7.966
Intercept	−4.659	0.729	40.859	1	<0.001	0.009		

**Table 5 microorganisms-12-00263-t005:** Association statistics for the multiple logistic regression between *A. hydrophila* occurrence and various factors. Refer to the legend in Table 2 for an explanation of statistical symbols.

Factor	B	S.E.	Wald	*d.f.*	*p*	Exp(B)	95% C.I.	
							Lower	Upper
Snake species (*B. kraepelini*)			21.561	5	<0.001			
* N. atra*	−1.842	0.531	12.047	1	<0.001	0.159	0.056	0.449
* B. multicinctus*	−0.140	0.435	0.103	1	0.748	0.870	0.371	2.040
* T. stejnegeri*	−1.286	0.561	5.257	1	0.022	0.276	0.092	0.830
* P. mucrosquamatus*	−0.722	0.531	1.849	1	0.174	0.486	0.172	1.375
* E. t. friesi*	−1.372	0.624	4.838	1	0.028	0.253	0.075	0.861
Presence of *E. faecalis*	1.055	0.397	7.073	1	0.008	2.871	1.320	6.244
Presence of *M. morganii*	1.257	0.278	20.495	1	<0.001	3.515	2.040	6.056
Presence of *P. aeruginosa*	1.307	0.301	18.872	1	<0.001	3.694	2.049	6.660

**Table 6 microorganisms-12-00263-t006:** A summary of factors significantly associated with the occurrence of four types of bacteria was determined through multiple logistic regression analyses. The reference category for subclass factors is indicated within parentheses following the specific factor name. Positive associations are denoted by ↑ (0.001 < *p* < 0.05) or ↑↑↑ (*p* < 0.001), while negative associations are denoted by ↓ (0.001 < *p* < 0.05) or ↓↓↓ (*p* < 0.001). For instance, the occurrence of *M. morganii* in two snake species (*N. atra* and *T. stejnegeri*) is significantly higher than in the reference species (*B. kraepelini*).

Factor	*E. faecalis*	*M. morganii*	*P. aeruginosa*	*A. hydrophila*
Snake species (*B. kraepelini*)				
* N. atra*	↓↓↓	↑↑↑		↓↓↓
* B. multicinctus*	↓			
* T. stejnegeri*	↓↓↓	↑↑↑		↓
* P. mucrosquamatus*	↓↓↓			
* E. t. friesi*				↓
Sex (Males)				
Females		↓		
Seasons (Winter 2021)				
Summer 2022			↓	
Temperature	↑↑↑	↓		
Elevation	(↑) *			
Coexisting bacteria	↑↑↑	↑↑↑	↑↑↑	↑↑↑

* The impact of elevation is minimal, as indicated by the fact that the B value for elevation is close to 0 (refer to Table 2).

## Data Availability

Data will be made available on request.

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
