# Peer review of "The Presence of Four Pathogenic Oral Bacterial Species in Six Wild Snake Species from Southern Taiwan: Associated Factors"

_microorganisms, 2024, doi:10.3390/microorganisms12020263_

Round 1

Reviewer 1 Report

Comments and Suggestions for Authors

The overall assessment

I enjoyed reading the manuscript. I found it valuable for publication, considering the authors performed an impressive study with extensive samples. However, there needs to be more clarification and explanation.

Questions

Figure 2: For easier understanding, you could include marker size on the gel picture. The picture remains to be improved.

Line 403: The authors assumed bacterial species may share growth environments concerning polybacterial oral conditions of the snakes. Were the snakes healthy and having appropriate immune activity? Are there any aspects explaining bacterial growth aside from venom?  

Line 423: As the authors understand, venom has a controversial role in the diversity of oral bacteria, a crucial point of this manuscript. You should incorporate more detail, compare, and address the possible hypothesis in your result.

Line 462: I am trying to figure out why you abruptly discussed the neurotoxic study. It needs to look more familiar to the overall context of the manuscript. You could explain its importance in the present study or remove it.

I found some grammatical errors as follows:

Line 74: The bacterial species carried the oral flora of snakes (à the bacterial species carried by the oral flora of snakes)

Line 103: Researches on these venomous snakes often involve (à Research on these venomous snakes often involves)

For the clarity of writing, I suggest the following:

Line 435: This observation aligns with findings from both clinical research on patient wounds and oral microbiota studies in snakes à this observation aligns with findings from clinical research on patient wounds and snake oral microbiota studies

Line 447: In this study, it exhibited à this study exhibited

Comments on the Quality of English Language

Fine. I only found a couple of issues. Overall, acceptable for publication.

Author Response

The overall assessment

I enjoyed reading the manuscript. I found it valuable for publication, considering the authors performed an impressive study with extensive samples. However, there needs to be more clarification and explanation.

Response: Thank you for providing valuable feedback. We have carefully considered your comments and have made the necessary adjustments to address the issues you raised.

Questions

Figure 2: For easier understanding, you could include marker size on the gel picture. The picture remains to be improved.

Response: We have incorporated marker size indicators on the gel picture and enhanced its clarity.

Line 403: The authors assumed bacterial species may share growth environments concerning polybacterial oral conditions of the snakes. Were the snakes healthy and having appropriate immune activity? Are there any aspects explaining bacterial growth aside from venom?  

Response: Factors influencing bacterial growth in a snake's oral cavity, aside from venom, encompass the equilibrium of oral microbiota, prey type, environmental conditions (humidity, temperature, and substrate composition), immune system function, oral hygiene, and the presence of injury or infection. In our study, we refrained from sampling snakes with compromised health or immune activity (Lines 156-157). Relevant discussions have been included at Line 406-409, 415-417.

Line 423: As the authors understand, venom has a controversial role in the diversity of oral bacteria, a crucial point of this manuscript. You should incorporate more detail, compare, and address the possible hypothesis in your result.

Response: We have provided additional details, conducted comparisons, and addressed potential hypotheses in the second paragraph of the Discussion section (Line 436-449).

Line 462: I am trying to figure out why you abruptly discussed the neurotoxic study. It needs to look more familiar to the overall context of the manuscript. You could explain its importance in the present study or remove it.

Response: We have eliminated the use of the term "neurotoxic" and rephrased the corresponding text in the passage located at Line 503-508.

I found some grammatical errors as follows:

Line 74: The bacterial species carried the oral flora of snakes (à the bacterial species carried by the oral flora of snakes)

Response: We have amended it accordingly (Line 83).

Line 103: Researches on these venomous snakes often involve (à Research on these venomous snakes often involves)

Response: We have amended it accordingly (Line 112).

For the clarity of writing, I suggest the following:

Line 435: This observation aligns with findings from both clinical research on patient wounds and oral microbiota studies in snakes à this observation aligns with findings from clinical research on patient wounds and snake oral microbiota studies

Response: We have amended it accordingly (Line 459).

Line 447: In this study, it exhibited à this study exhibited

Response: We have amended it accordingly (Line 467).

Comments on the Quality of English Language

Fine. I only found a couple of issues. Overall, acceptable for publication.

Response: Your feedback is appreciated. We are grateful for your positive assessment. Thank you.

Reviewer 2 Report

Comments and Suggestions for Authors

Manuskript-ID:   microorganisms_2825123

Journal:               Microorganisms

Title:                     The presence of four pathogenic oral bacterial species in six  wild snake species from Southern Taiwan: Associated factors Authors:              Wen-Hao Lin, Tein-Shun Tsai, and Po-Chun Chuang

Short Summary

The authors submitted a manuscript that describes the examination of six Taiwanese snake species, both venomous and non-venomous ones, for the occurrence of four different bacterial species within the oral cavities of the snake individuals. Sophisticated statistical analysis revealed that several factors including species, gender, temperature, season, and coexisted pathogens may have an impact on the occurrence of the target bacteria, whereas others including body size parameters, sampling site characteristics and weather conditions like rainfall do not.

General Impression

The manuscript addresses a topic that is of potential interest for a broad audience: the public, health authorities , medical stuff and the scientific community in venomous animal research. The authors used established techniques in molecular genotyping as well as comprehensive tools in statistical data analysis and describe in great detail the results of their investigations. I greatly appreciate all the efforts and have no doubts that the experimental and analytical work was properly done. The manuscript is well written and it was a pleasure to read it. I have only a few remarks that can be easily addressed.  

Minor concerns:

1) In the light of the 2023 publication in Biology (Basel) (reference 16) I recommend to clarify early in the Introduction chapter why the targeted approach was used (instead a non-targeted approach) and to  better justify the selection of the four bacterial species.

2) Table 6 is a little difficult to understand. It should be clarified that the associations that are indicated by the arrows describe the differences between the conditions in a certain snake species (e.g. N. atra) and B. kraepelini as the reference species. If this is not the case than I misunderstood the figure by myself.

3) The indicated primer positions in chapter 2.3 are incorrect. Here are the correct positions:

                E. faecalis:          167 - 189 (fw) and 459 - 476 (rev)

                M. morganii:     357 - 376 (fw) and 515 - 534 (rev)

                P. aeruginosa:  185 - 202 (fw) and 1122 - 1140 (rev)

                A. hydrophila:   445 - 467 (fw) and 1110 - 1130 (rev)

4) The size of the expected amplicon in M. morganii is 178 bp, not 179 bp (lane 198 and legend to Figure 2).   

5) lanes 354 - 357: The species names are not written in italic, please correct.

6) lane 24: "banding results" sounds strange, please rephrase

7) lane 38-39: " 81,000 and 138,000 " are mortality cases, not a mortality rate (rate is cases per 100,000)

Author Response

Manuskript-ID:   microorganisms_2825123

Journal:               Microorganisms

Title:                     The presence of four pathogenic oral bacterial species in six  wild snake species from Southern Taiwan: Associated factors

Authors:              Wen-Hao Lin, Tein-Shun Tsai, and Po-Chun Chuang

Short Summary

The authors submitted a manuscript that describes the examination of six Taiwanese snake species, both venomous and non-venomous ones, for the occurrence of four different bacterial species within the oral cavities of the snake individuals. Sophisticated statistical analysis revealed that several factors including species, gender, temperature, season, and coexisted pathogens may have an impact on the occurrence of the target bacteria, whereas others including body size parameters, sampling site characteristics and weather conditions like rainfall do not.

General Impression

The manuscript addresses a topic that is of potential interest for a broad audience: the public, health authorities , medical stuff and the scientific community in venomous animal research. The authors used established techniques in molecular genotyping as well as comprehensive tools in statistical data analysis and describe in great detail the results of their investigations. I greatly appreciate all the efforts and have no doubts that the experimental and analytical work was properly done. The manuscript is well written and it was a pleasure to read it. I have only a few remarks that can be easily addressed.  

Response: Thank you for providing valuable feedback. We have carefully considered your comments and have made the necessary adjustments to address the issues you raised.

Minor concerns:

1) In the light of the 2023 publication in Biology (Basel) (reference 16) I recommend to clarify early in the Introduction chapter why the targeted approach was used (instead a non-targeted approach) and to  better justify the selection of the four bacterial species.

Response: We have added more explanation in the third paragraph of Introduction (Line 58-82).

2) Table 6 is a little difficult to understand. It should be clarified that the associations that are indicated by the arrows describe the differences between the conditions in a certain snake species (e.g. N. atra) and B. kraepelini as the reference species. If this is not the case than I misunderstood the figure by myself.

Response: We have added more explanation at Line 432-433.

3) The indicated primer positions in chapter 2.3 are incorrect. Here are the correct positions:

  1. faecalis:          167 - 189 (fw) and 459 - 476 (rev)
  2. morganii:     357 - 376 (fw) and 515 - 534 (rev)
  3. aeruginosa:  185 - 202 (fw) and 1122 - 1140 (rev)
  4. hydrophila:   445 - 467 (fw) and 1110 - 1130 (rev)

Response: Thank you for showing the correct positions (the fw primer position for P. aeruginosa should be 185-203) and we have amended them accordingly (Line 185, 187, 201, 203, 211, 213, 221, 223).

4) The size of the expected amplicon in M. morganii is 178 bp, not 179 bp (lane 198 and legend to Figure 2).   

Response: Thank you for showing the correct size and we have amended them accordingly (Line 197, 207).

5) lanes 354 - 357: The species names are not written in italic, please correct.

Response: We have amended them (Line 363-366).

6) lane 24: "banding results" sounds strange, please rephrase

Response: We have rephrased it (Line 24).

7) lane 38-39: " 81,000 and 138,000 " are mortality cases, not a mortality rate (rate is cases per 100,000)

Response: We have amended it (Line 38).
